# What Can We Learn from the Past by Means of Very Long-Term Follow-Up after Aortic Valve Replacement?

**DOI:** 10.3390/jcm10173925

**Published:** 2021-08-31

**Authors:** Ben Swinkels, Jurriën ten Berg, Johannes Kelder, Freddy Vermeulen, Wim Jan van Boven, Bas de Mol

**Affiliations:** 1Department of Cardiology, St. Antonius Hospital, 3435 CM Nieuwegein, The Netherlands; j.ten.berg@antoniusziekenhuis.nl (J.t.B.); keld01@antoniusziekenhuis.nl (J.K.); 2Department of Cardiothoracic Surgery, St. Antonius Hospital, 3435 CM Nieuwegein, The Netherlands; freddy.vermeulen@outlook.be; 3Academic Medical Center, Department of Cardiothoracic Surgery, Amsterdam University Medical Center, 1105 AZ Amsterdam, The Netherlands; w.j.vanboven@amsterdamumc.nl (W.J.v.B.); b.a.demol@amsterdamumc.nl (B.d.M.)

**Keywords:** valvular heart surgery, aortic valve replacement, long-term follow-up, survival analysis

## Abstract

Background: Studies on very long-term outcomes after aortic valve replacement are sparse. Methods: In this retrospective cohort study, long-term outcomes during 25.1 ± 2.8 years of follow-up were determined in 673 patients who underwent aortic valve replacement with or without concomitant coronary artery bypass surgery for severe aortic stenosis and/or regurgitation. Independent predictors of decreased long-term survival were determined. Cumulative incidence rates of major adverse events in patients with a mechanical versus those with a biologic prosthesis were assessed, as well as of major bleeding events in patients with a mechanical prosthesis under the age of 60 versus those above the age of 60. Results: Impaired left ventricular function, severe prosthesis–patient mismatch, and increased aortic cross-clamp time were independent predictors of decreased long-term survival. Left ventricular hypertrophy, a mechanical or biologic prosthesis, increased cardiopulmonary bypass time, new-onset postoperative atrial fibrillation, and the presence of symptoms did not independently predict decreased long-term survival. The risk of major bleeding events was higher in patients with a mechanical in comparison with those with a biologic prosthesis. Younger age (under 60 years) did not protect patients with a mechanical prosthesis against major bleeding events. Conclusions: Very long-term outcome data are invaluable for careful decision-making on aortic valve replacement.

## 1. Introduction

Publications on diagnosis and treatment of valvular heart disease tend to focus on new technologies [1,2,3,4]. When patients undergo cardiac surgery for heart valve replacement, the implanted prosthesis does have a lifetime impact [5,6]. Advancements in devices and disease management are often embraced on the basis of relatively short-term results [7,8]. However, the final test of whether and to what degree new heart valve treatments are indeed an improvement in life expectancy and quality of life is determined by comparisons of long-term and very long-term outcomes [9,10,11]. In addition, we know that loss of lives, loss of life expectancy, and quality of life are hard to repair and are determined greatly by the quality of care in a particular hospital [12,13,14]. Parameters for the quality of an institution’s practice include the incidence of postoperative ischemic or hemorrhagic stroke, need for a permanent pacemaker implantation, readmission rate, duration of aortic cross-clamp (ACC) and cardiopulmonary bypass (CPB) time, occurrence of new-onset postoperative atrial fibrillation (POAF), and others [15,16,17,18,19,20,21,22,23,24,25]. Long-term survival after aortic valve replacement (AVR) can be affected by patient-associated preoperative risk factors, such as impaired left ventricular (LV) function, previous coronary artery bypass grafting (CABG), hypertension, chronic lung disease, previous ischemic stroke, peripheral arterial disease, diabetes, atrial fibrillation, high or low body mass index, previous myocardial infarction, renal impairment, or severe pulmonary hypertension [26,27,28,29,30]. In the late 1980s, the question arose of how to improve long-term surgical outcomes, but also on how surgical techniques may have an impact on long-term quality of life, at the department of Cardiothoracic Surgery of the St. Antonius Hospital at Nieuwegein, which is a large volume cardiothoracic center in The Netherlands with 1791 open-heart surgical procedures in 2019 [31]. This resulted in the “Survival and QUality of life after Aortic valve REplacement Surgery" (SQUARES) project. Initially, the main purpose of the project was to study long-term outcomes after AVR, either isolated or with concomitant CABG, as well as to collect quality of life data during long-term follow-up on a regular basis of at least 10 years. Fortunately, the registry enabled regular further follow-up of at least 20 years, which provided the basis for the results of the current study. In the current, retrospective cohort study, we aimed to determine the effect of impaired LV function, LV hypertrophy (LVH), moderate or severe prosthesis–patient mismatch (PPM), a mechanical or biologic prosthesis, ACC time, CPB time, new-onset POAF, and the presence of symptoms, on long-term survival after AVR with or without concomitant CABG for symptomatic or asymptomatic severe aortic stenosis (AS) and/or aortic regurgitation (AR). We also aimed to determine cumulative incidence rates of major adverse valve-related events in patients with a mechanical versus (vs.) those with a biologic prosthesis, as well as cumulative incidence rates of major bleeding events during long-term follow-up in patients with a mechanical prosthesis under vs. those above the age of 60.

## 2. Materials and Methods

### 2.1. Study Population

The study population comprised a cohort of 673 consecutive patients with symptomatic or asymptomatic severe AS and/or AR, who underwent AVR, either isolated or with concomitant CABG, between 1990 and 1994 in a single center. Exclusion criteria were repeat AVR, aortic homograft surgery, and AVR combined with ascending aortic graft replacement surgery, ventricular septal defect, LV aneurysm resection, lung surgery, and/or other heart valve surgery. The effect on long-term survival of impaired LV function, LVH, moderate PPM, severe PPM, mechanical prosthesis, biologic prosthesis, ACC time, CPB time, new-onset POAF, and the presence of symptoms was determined in different subgroups of patients with different subgroup-specific baseline characteristics, depending on the research question.

The effect of impaired LV function on long-term survival was determined in a subgroup of 404 consecutive patients, divided into 62 patients with impaired LV function (LV ejection fraction (LVEF) < 50%) and 342 patients with preserved LV function (LVEF ≥ 50%), who underwent isolated AVR for symptomatic severe AS. To assess the effect of impaired LV function due to severe AS, instead of due to a permanently impaired LV function because of previous myocardial infarction or nonfatal perioperative myocardial infarction, the effect of impaired LV function on long-term survival was also determined in patients without previous myocardial infarction or nonfatal perioperative myocardial infarction.

The effect of LVH on long-term survival was determined in a subgroup of 404 consecutive patients, divided into 300 patients with LVH and 104 patients without LVH, who underwent isolated AVR for symptomatic severe AS. To assess the effect on long-term survival of LVH due to severe AS, instead of due to hypertension, the effect of LVH on long-term survival was also determined in patients without a history of hypertension.

The effect of PPM on long-term survival was determined in the total study population of 673 consecutive patients, divided into 163 patients with PPM (subdivided into 146 patients with moderate and 17 patients with severe PPM), and 510 patients with no/mild PPM, who underwent AVR with or without concomitant CABG for symptomatic or asymptomatic severe AS and/or AR. Moderate, severe, and no/mild PPM were defined as indexed effective orifice area of >0.65, ≤0.85, ≤0.65, and >0.85 cm^2^/m^2^, respectively.

The effect of a mechanical or biologic prosthesis on long-term survival was determined in the total study population of 673 consecutive patients, divided into 430 patients with a mechanical and 243 patients with a biologic prosthesis, who underwent AVR with or without concomitant CABG for symptomatic or asymptomatic severe AS and/or AR.

The effect of ACC time on long-term survival was determined in a subgroup of 670 consecutive patients (673 patients of the total study population excluding three patients with unknown ACC and/or CPB times), divided into 334 patients with ACC time of less than 68.0 min and 336 patients with ACC time of 68.0 min or more, who underwent AVR with or without concomitant CABG for symptomatic or asymptomatic severe AS and/or AR. The cutoff value of 68.0 min was arbitrarily based on the median ACC time of 68.0 min to create two similarly sized groups. To assess the effect of ACC time on long-term survival independently of CPB time, both the variables ACC time and CPB time were included in the multivariable analysis. Because a concomitant enlargement procedure lengthens ACC time, the variable concomitant enlargement procedure was also included in the multivariable analysis.

The effect of CPB time on long-term survival was determined in the same subgroup of 670 consecutive patients as the subgroup in which the effect of ACC time on long-term survival was determined, divided into 331 patients with CPB time of less than 95.0 min and 339 patients with CPB time of 95.0 min or more, who underwent AVR with or without concomitant CABG for symptomatic or asymptomatic severe AS and/or AR. The cutoff value of 95.0 min was arbitrarily based on the median CPB time of 95.0 min to create two similarly sized groups. To assess the effect of CPB time on long-term survival independently of ACC time, both the variables CPB time and ACC time were included in the multivariable analysis. Because a concomitant enlargement procedure lengthens CPB time, the variable concomitant enlargement procedure was also included in the multivariable analysis.

The effect of new-onset POAF on long-term survival was determined in a subgroup of 569 consecutive patients, divided into 241 patients with new-onset POAF and 328 patients without new-onset POAF, who underwent AVR with or without concomitant CABG for symptomatic or asymptomatic severe AS and/or AR. New-onset POAF was defined as de novo atrial fibrillation in patients without a history of atrial fibrillation, lasting for at least several hours, and occurring postoperatively when the patient was still admitted. Treatment of new-onset POAF was aimed to restore sinus rhythm within 24 to 48 h from the onset by use of medication and/or direct-current cardioversion before discharge from hospital. Patients with new-onset POAF who were still in atrial fibrillation at discharge underwent direct-current cardioversion within 6 weeks from discharge.

The effect of the presence of symptoms on long-term survival was determined in a subgroup of 562 consecutive patients, divided into 50 asymptomatic and 512 symptomatic patients, who underwent AVR with or without concomitant CABG for severe AS. Asymptomatic patients were defined as those who had been in New York Heart Association (NYHA) Class I and symptomatic patients as those who had been in NYHA Class II–IV.

Cumulative incidence rates of major adverse valve-related events, comprising major bleeding, hemorrhagic stroke, ischemic stroke, prosthetic valve endocarditis, permanent pacemaker implantation, and repeat AVR, during long-term follow-up after discharge from hospital in patients with a mechanical vs. those with a biologic prosthesis were determined in the total study population of 673 patients, divided into 430 patients with a mechanical and 243 patients with a biologic prosthesis, who underwent AVR with or without concomitant CABG for symptomatic or asymptomatic severe AS and/or AR.

Cumulative incidence rates of major mechanical valve-related bleeding events during long-term follow-up after discharge from hospital in patients with a mechanical prosthesis under vs. those above the age of 60 were determined in a subgroup of 430 patients with a mechanical prosthesis, divided into 163 patients under the age of 60 and 267 patients aged 60 years or more at the time of surgery, who underwent AVR with or without concomitant CABG for symptomatic or asymptomatic severe AS and/or AR.

Follow-up data were obtained from our own or referring cardiology departments, general practitioners, and telephone calls to patients and relatives. Target international normalized ratio (INR) values during follow-up were 2.5 to 4.0 for the patients with a mechanical prosthesis and 2.0 to 3.5 for those with a biologic prosthesis and an indication for oral anticoagulation therapy such as atrial fibrillation. These target INR values were the standard national target INR values for these patients at the time in the Netherlands [32]. In 2016, both of these target INR values were lowered to the international target INR values of 2.5 to 3.5 and 2.0 to 3.0, respectively [33].

### 2.2. Baseline Characteristics

Baseline characteristics were defined as follows. LVH: interventricular septal thickness of at least 11 mm. Recent or old myocardial infarction: myocardial infarction occurring within 6 or more than 6 weeks before AVR, respectively. Previous ischemic stroke: focal neurological deficit of sudden onset as diagnosed by a neurologist, lasting more than 24 h and caused by cerebral ischemia. Peripheral arterial disease: claudication, surgical, or percutaneous intervention on the peripheral arteries, excluding carotid disease. Hypertension: blood pressure of more than 140/90 mmHg or use of antihypertensive medication. Severe pulmonary hypertension: mean invasive pulmonary artery pressure of more than 40 mmHg. Chronic lung disease: chronic obstructive lung disease, asthma, or pulmonary fibrosis. Presence or absence of symptoms: NYHA Class II-IV or I, respectively. Peak gradient across the aortic valve, aortic valve area, LVEF, and interventricular septal thickness were all measured by transthoracic echocardiography.

### 2.3. Candidate Independent Predictors of Decreased Late Survival

A priori based on previous research [28,29,30], the following candidate independent predictors of decreased long-term survival were included in the multivariable analysis: age (continuous variable), predominant AS, predominant AR, concomitant CABG, previous CABG, hypertension, chronic lung disease, previous ischemic stroke, peripheral arterial disease, insulin-dependent diabetes, non-insulin-dependent diabetes, paroxysmal atrial fibrillation, permanent atrial fibrillation, body mass index (continuous variable), recent myocardial infarction, old myocardial infarction, serum creatinine (continuous variable), and severe pulmonary hypertension. Depending on the different research questions of the current study, the following candidate independent predictors were also included: impaired LV function, LVH, moderate PPM, severe PPM, mechanical prosthesis, biologic prosthesis, ACC time (continuous variable), CPB time (continuous variable), concomitant annular enlargement, new-onset POAF, and the presence of symptoms.

### 2.4. Statistical Analysis

Kaplan–Meier survival analysis was performed to determine the long-term survival, from date of surgery, of patients with the following preoperative or operative characteristics: impaired vs. preserved LV function, LVH vs. no LVH, moderate vs. no/mild PPM, severe vs. no/mild PPM, moderate vs. severe PPM, ACC time of less than 68.0 min vs. ACC time of 68.0 min or more, CPB time of less than 95.0 min vs. CPB time of 95.0 min or more, mechanical vs. biologic prosthesis, new-onset POAF vs. no new-onset POAF, and the presence of symptoms vs. the absence of symptoms. Cox proportional hazards analysis was performed to determine the effect on long-term survival of the following candidate independent predictors of decreased long-term survival, after excluding the patients who had died within 30 days from surgery: impaired LV function, LVH, moderate PPM, severe PPM, mechanical prosthesis, biologic prosthesis, ACC time (continuous variable), CPB time (continuous variable), new-onset POAF, and the presence of symptoms. Binary variables were labeled as yes vs. no or missing. Missing data on categorical variables were automatically categorized into the lowest-risk category. Continuous outcomes and dichotomous variables were analyzed using Student’s *t*-tests and chi-square tests, respectively. Two-sided tests of significance are reported, and *p* values <0.05 were considered statistically significant. Data were analyzed by SPSS Statistics 25.

## 3. Results

Mean long-term follow-up, from date of surgery, of the 673 patients of the total study population was 25.1 ± 2.8 years. Follow-up was complete in all but 11 (1.6%) patients, who were lost to follow-up at 16 (*n* = 2), 17 (*n* = 2), 18 (*n* = 3), 19 (*n* = 3), and 20 (*n* = 1) years after surgery.

### 3.1. Impaired LV Function

The subgroup of patients in which the effect of impaired LV function on long-term survival was assessed comprised 404 patients, divided into 62 patients with impaired and 342 patients with preserved LV function, who underwent isolated AVR for symptomatic severe AS.

#### 3.1.1. Baseline Characteristics

As shown in Table 1, the mean age of the patients with impaired vs. those with preserved LV function was similar. However, operative risk scores in patients with impaired LV function were higher.

#### 3.1.2. Thirty-Day Mortality

Thirty-day mortality was similar in the patients with impaired vs. those with preserved LV function: 0.0% (*n* = 0) vs. 2.6% (*n* = 9), respectively; *p* = 0.220.

#### 3.1.3. Long-Term Survival

As shown in Figure 1, long-term survival was worse in patients with impaired LV function in comparison with those with preserved LV function: mean 10.8 ± 0.9 (95% confidence interval (CI): 9.0–12.7) vs. 14.0 ± 0.5 (95% CI: 13.0–14.9) years, respectively; log-rank: *p* = 0.004.

#### 3.1.4. Multivariable Analysis

In Cox proportional hazards analysis, impaired LV function was an independent predictor of decreased long-term survival: hazards ratio (HR) 1.563 (95% CI: 1.124−2.174); *p* = 0.008. After excluding all patients with a permanently impaired LV function due to previous myocardial infarction (19 patients with preserved and 10 patients with impaired LV function), or due to nonfatal perioperative myocardial infarction (two patients with preserved and zero patients with impaired LV function), impaired LV function was still an independent predictor of decreased long-term survival: HR 1.446 (95% CI: 1.019−2.052); *p* = 0.039.

#### 3.1.5. Time Course of Impaired LV Function

Figure 2 shows the time course of impaired LV function during 10 years of follow-up. Echocardiographic data on LV function beyond 10 years of follow-up were not available. LV function had improved from impaired into preserved in 39 of 60 (65.0%) surviving patients at 1 year of follow-up, being still impaired in 15 of 60 (25.0%) surviving patients at 1 year of follow-up. However, LV function had deteriorated again into being impaired in 19 of 32 (59.4%) surviving patients at 10 years of follow-up.

#### 3.1.6. Conclusions

Impaired LV function was an independent predictor of decreased long-term survival after isolated AVR in patients with symptomatic severe AS. Earlier surgery, before the development of impaired LV function, may therefore improve long-term survival.

### 3.2. LVH

The subgroup of patients in which the effect of LVH on long-term survival was assessed comprised 404 patients, divided into 300 patients with LVH and 104 patients without LVH, who underwent isolated AVR for symptomatic severe AS.

#### 3.2.1. Baseline Characteristics

As shown in Table 2, patients with LVH were older in comparison with those without LVH. However, operative risk scores were similar.

#### 3.2.2. Thirty-Day Mortality

Thirty-day mortality was similar in the patients with LVH vs. those without LVH: 2.7% (*n* = 8) vs. 1.0% (*n* = 1), respectively; *p* = 0.457.

#### 3.2.3. Long-Term Survival

As shown in Figure 3, long-term survival was similar in patients with vs. those without LVH: mean 13.3 ± 0.5 (95% CI: 12.4–14.3) vs. 13.9 ± 0.9 (95% CI: 12.1–15.7) years, respectively; log-rank: *p* = 0.296.

#### 3.2.4. Multivariable Analysis

In Cox proportional hazards analysis, LVH was not an independent predictor of decreased long-term survival either in the total study population or in the 218 patients without a history of hypertension.

#### 3.2.5. Time Course of LVH

Figure 4 shows the time course of LVH during 10 years of follow-up, after excluding the patients with a history of hypertension. Echocardiographic data on LVH beyond 10 years of follow-up were not available. LVH had substantially regressed at 1 year of follow-up, being still present in one-half of surviving patients at 1 year of follow-up, followed by further regression to less than half of surviving patients at 10 years of follow-up.

#### 3.2.6. Conclusions

LVH due to severe AS was not associated with decreased long-term survival after isolated AVR in patients with symptomatic severe AS. Therefore, LVH per se is not an indication for AVR in patients with symptomatic severe AS. This finding conforms to current American and European guidelines for the management of valvular heart disease [34,35].

### 3.3. PPM

The effect of PPM on long-term survival was assessed in the total study population of 673 patients, divided into 163 patients with PPM (subdivided into 146 patients with moderate and 17 patients with severe PPM) and 510 patients with no/mild PPM, who underwent AVR with or without concomitant CABG for symptomatic or asymptomatic severe AS and/or AR.

PPM occurred in 88 of 430 (20.5%) patients with a mechanical prosthesis and in 75 of 243 (30.9%) patients with a biologic prosthesis. Severe PPM occurred in 8 of 430 (1.9%) patients with a mechanical and in 9 of 243 (3.3%) patients with a biologic prosthesis. PPM occurred only in patients with a mechanical prosthesis sized 23 mm or smaller or with a biologic prosthesis sized 25 mm or smaller. Severe PPM occurred only in patients with a mechanical prosthesis sized 23 mm or smaller or with a biologic prosthesis sized 23 mm.

#### 3.3.1. Baseline Characteristics

As shown in Table 3, patients with PPM were older in comparison with those with no/mild PPM. Operative risk scores in patients with PPM showed a non-significant trend toward being higher in comparison with those with no/mild PPM.

After subdividing the patients with PPM into patients with moderate and those with severe PPM, the mean ages of the patients with severe PPM vs. those with moderate PPM were similar, as were their operative risk scores. This is shown in Table 4.

#### 3.3.2. Thirty-Day Mortality

Thirty-day mortality was similar in the patients with PPM vs. those with no/mild PPM: 1.2% (*n* = 2) vs. 3.3% (*n* = 17), respectively; *p* = 0.274. Thirty-day mortality was also similar in the patients with severe PPM vs. those with moderate PPM: 0.0% (*n* = 0) vs. 1.4% (*n* = 2), respectively; *p* = 0.802.

#### 3.3.3. Long-Term Survival

Long-term survival of the patients with moderate PPM, severe PPM, and no/mild PPM was 12.7 ± 0.7 (95% CI: 11.5–14.0), 8.6 ± 2.1 (95% CI: 4.4–12.8), and 13.1 ± 0.4 (95% CI: 12.4–13.9) years, respectively. As shown in Figure 5, long-term survival showed a non-significant trend toward being worse in patients with severe PPM in comparison with those with no/mild PPM (log-rank: *p* = 0.052), was similar in patients with severe PPM vs. those with moderate PPM (log-rank: *p* = 0.191) and was also similar in patients with moderate PPM vs. those with no/mild PPM (log-rank: *p* = 0.314).

#### 3.3.4. Multivariable Analysis

In Cox proportional hazards analysis, severe PPM was an independent predictor of decreased long-term survival: HR 2.247 (95% CI: 1.241–4.070); *p* = 0.008. Moderate PPM was not an independent predictor of decreased long-term survival: HR 0.856 (95% CI: 0.685–1.071); *p* = 0.174. When the patients who had died within 30 days from surgery were not excluded but were included, like in a previous study with the same study population as the current study, but with a shorter long-term follow-up (mean 17.8 ± 1.8 years), in which patients with severe PPM had only shown a non-significant trend toward decreased long-term survival [36], severe PPM was still an independent predictor of decreased long-term survival (HR 2.001 (95% CI: 1.106–3.618); *p* = 0.022), while moderate PPM did still not independently predict decreased long-term survival (HR 0.839 (95% CI: 0.672–1.046); *p* = 0.199).

The finding that moderate PPM was not an independent predictor of decreased long-term survival and even showed a non-significant trend toward an increased, not a decreased, long-term survival in comparison with no/mild PPM, was difficult to explain because patients with moderate PPM had not been younger and their operative risk scores had not been lower in comparison with those with no/mild PPM. This inverse survival relationship between moderate PPM and long-term survival was also found in our aforementioned previous paper on PPM [36] and, as described in that paper, could also not be explained statistically. So was the robustness of the used Cox regression model at least reasonable and after putting other variables into the model, such as STS-PROM operative risk score, this inverse survival relationship was not essentially changed. Additionally, using the indexed effective orifice area as a continuous variable, both linearly and nonlinearly, or stratifying by age group, or using age as a nonlinear variable, did not essentially change this inverse survival relationship. Looking only at the first year of follow-up, this inverse survival relationship was also retained.

#### 3.3.5. Conclusions

Severe, but not moderate, PPM was an independent predictor of decreased long-term survival after AVR with or without concomitant CABG for symptomatic or asymptomatic severe AS and/or AR. Therefore, severe PPM should be prevented as much as possible.

### 3.4. Mechanical vs. Biologic Prosthesis

The effect on long-term survival of patients with a mechanical vs. those with a biologic prosthesis was assessed in the total study population of 673 patients, divided into 430 patients with a mechanical and 243 patients with a biologic prosthesis, who underwent AVR with or without concomitant CABG for symptomatic or asymptomatic severe AS and/or AR.

The following mechanical (*n* = 430) and biologic (*n* = 243) prostheses were implanted. Mechanical prostheses: St. Jude Medical Standard (*n* = 205), St. Jude Medical Hemodynamic Plus (*n* = 3), Sorin Allcarbon (*n* = 211, including two 33 mm sized Sorin Allcarbon mitral valve mechanical prostheses, placed upside down in the aortic position), and the Sorin Bicarbon mechanical prosthesis (*n* = 11). Biologic prostheses: Medtronic Intact (*n* = 100), CE-SAV (*n* = 139), and CE Perimount biologic prosthesis (*n* = 4). The most frequent prosthesis size among the mechanical prostheses was 23 mm (*n* = 128), followed by 25 mm (*n* = 113). The most frequent prosthesis size among the biologic prostheses was also 23 mm (*n* = 101), followed by 25 mm (*n* = 80).

#### 3.4.1. Baseline Characteristics

As shown in Table 5, patients with a mechanical prosthesis were much younger in comparison with those with a biologic prosthesis. Operative risk scores in the patients with a mechanical prosthesis were lower in comparison with those with a biologic prosthesis.

That patients with a mechanical prosthesis were much younger than those with a biologic prosthesis is illustrated in the histogram of Figure 6, showing that almost all patients who had been under the age of 68 at the time of surgery had a mechanical prosthesis, while almost all patients who had been above the age of 74 at the time of surgery had a biologic prosthesis.

#### 3.4.2. Thirty-Day Mortality

Thirty-day mortality was higher in the patients with a biologic in comparison with those with a mechanical prosthesis: 4.5% (*n* = 11) vs. 1.9% (*n* = 8), respectively; *p* = 0.041.

#### 3.4.3. Long-Term Survival of Patients with a Mechanical vs. Those with a Biologic Prosthesis

As shown in Figure 7, long-term survival was worse in patients with a biologic vs. those with a mechanical prosthesis: 9.0 ± 0.4 (95% CI: 8.3–9.7) vs. 15.2 ± 0.4 (95% CI: 14.3–16.0) years, respectively; log-rank: *p* < 0.001.

#### 3.4.4. Multivariable Analysis

In the Cox proportional hazards analysis, neither a biologic nor a mechanical prosthesis was an independent predictor of decreased long-term survival.

#### 3.4.5. Conclusions

Although long-term survival after AVR with or without concomitant CABG for symptomatic or asymptomatic severe AS and/or AR of the much younger patients with a mechanical prosthesis was better in comparison with those with a biologic prosthesis, after multivariable analysis including age, neither a mechanical nor a biologic prosthesis was an independent predictor of decreased long-term survival. However, because both the numbers of patients with a biologic prosthesis under the age of 68 and those with a mechanical prosthesis above the age of 74 were so small, it cannot be concluded from the findings of the current study that mechanical and biologic prostheses are equally safe in both younger and older patients regarding long-term survival.

### 3.5. ACC Time

The subgroup of patients in which the effect of ACC time on long-term survival was assessed comprised 670 patients, divided into 334 patients with ACC time of less than 68.0 min and 336 patients with ACC time of 68.0 min or more, who underwent AVR with or without concomitant CABG for symptomatic or asymptomatic severe AS and/or AR.

#### 3.5.1. Baseline Characteristics

As shown in Table 6, the mean age of the patients with ACC time of 68.0 min or more showed a non-significant trend to be higher in comparison with those with an ACC time of less than 68.0 min. Operative risk scores were higher in the patients with ACC time of 68.0 min or more in comparison with those with ACC time of less than 68.0 min.

#### 3.5.2. Thirty-Day Mortality

Thirty-day mortality was similar in the patients with ACC time of 68.0 min or more vs. those with ACC time of less than 68.0 min: 3.9% (*n* = 13) vs. 1.8% (*n* = 6), respectively; *p* = 0.161.

#### 3.5.3. Long-Term Survival

As shown in Figure 8, long-term survival in patients with ACC time of 68.0 min or more was worse in comparison with those with ACC time of less than 68.0 min: 11.6 ± 0.4 years (95% CI: 10.7–12.4) vs. 14.3 ± 0.5 years (95% CI: 13.4–15.2), respectively; log-rank: *p* < 0.001.

#### 3.5.4. Multivariable Analysis

In Cox proportional hazards analysis, increased ACC time, even within normal limits, was an independent predictor of decreased long-term survival: per minute increase of ACC time, the HR for decreased late survival was 1.006 (95% CI: 1.002–1.009; *p* = 0.002). The finding that increased ACC time, even within normal limits, was an independent predictor of decreased long-term survival was also done in a previous study with the same long-term follow-up (mean 25.3 ± 2.7 years) as in the current study, but with a different study population comprising a subpopulation of 456 consecutive patients of the current study who underwent isolated aortic valve replacement for symptomatic or asymptomatic severe AS [37].

#### 3.5.5. Conclusions

Increased ACC time, even within normal limits, was an independent predictor of decreased long-term survival after AVR with or without concomitant CABG for symptomatic or asymptomatic severe AS and/or AR. Therefore, ACC time should be kept as short as possible.

### 3.6. CPB Time

The subgroup of patients in which the effect of CPB time on long-term survival was assessed comprised 331 patients with CPB time of less than 95.0 min and 339 patients with CPB time of 95.0 min or more, who underwent AVR with or without concomitant CABG for symptomatic or asymptomatic severe AS and/or AR.

#### 3.6.1. Baseline Characteristics

As shown in Table 7, the mean age of the patients with CPB time of 95.0 min or more was higher, as were their operative risk scores, in comparison with those with CPB time of less than 95.0 min.

#### 3.6.2. Thirty-Day Mortality

Thirty-day mortality showed a non-significant trend to be higher in the patients with CPB time of 95.0 min or more, in comparison with those with CPB time of less than 95.0 min: 4.1% (*n* = 14) vs. 1.5% (*n* = 5), respectively; *p* = 0.060.

#### 3.6.3. Long-Term Survival

As shown in Figure 9, long-term survival in patients with CPB time of 95.0 min or more was worse in comparison with those with CPB time of less than 95.0 min: 11.8 ± 0.5 years (95% CI: 10.9–12.7) vs. 14.1 ± 0.5 years (95% CI: 13.1–15.0), respectively; log-rank: *p* = 0.001.

#### 3.6.4. Multivariable Analysis

In Cox proportional hazards analysis, increased CPB time was not an independent predictor of decreased long-term survival. This finding was also done in a previous study with the same long-term follow-up (mean 25.3 ± 2.7 years) as in the current study, but with a different study population comprising a subpopulation of 456 consecutive patients of the current study who underwent isolated aortic valve replacement for symptomatic or asymptomatic severe AS [37].

#### 3.6.5. Conclusions

Increased CPB time was associated with both higher age and higher operative risk scores in patients who underwent AVR with or without concomitant CABG for symptomatic or asymptomatic severe AS and/or AR. However, increased CPB time was not an independent predictor of decreased long-term survival. Therefore, CPB time seems to be inherently increased in higher-risk surgery due to higher age and/or more complicated surgery, and reduction of CPB time per se does not seem to improve long-term survival when age and operative risk scores are left unchanged.

### 3.7. New-Onset POAF

The subgroup of patients in which the effect of new-onset POAF on long-term survival was assessed comprised 569 patients, divided into 241 patients with new-onset POAF and 328 patients without new-onset POAF, who underwent AVR with or without concomitant CABG for symptomatic or asymptomatic severe AS and/or AR. Direct-current cardioversion within 6 weeks from discharge from hospital in the patients with new-onset POAF who were still in atrial fibrillation at discharge from hospital was successful in the vast majority of cases.

#### 3.7.1. Baseline Characteristics

As shown in Table 8, the mean ages of the patients with new-onset POAF vs. those without new-onset POAF were similar, as were their operative risk scores.

#### 3.7.2. Thirty-Day Mortality

Thirty-day mortality was similar in the patients with new-onset POAF vs. those without new-onset POAF: 1.2% (*n* = 3) vs. 2.7% (*n* = 9), respectively; *p* = 0.254.

#### 3.7.3. Long-Term Survival

As shown in Figure 10, long-term survival was similar in patients with vs. those without new-onset POAF: 13.8 ± 0.5 (95% CI: 12.8–14.9) vs. 13.3 ± 0.5 (95% CI: 12.4–14.3) years, respectively; log-rank: *p* = 0.867.

#### 3.7.4. Multivariable Analysis

In Cox proportional hazards analysis, new-onset POAF was not an independent predictor of decreased long-term survival. This finding was also done in a previous study with the same study population as the current study, but with a shorter long-term follow-up (mean 17.8 ± 1.8 years) than in the current study [38].

#### 3.7.5. Conclusions

New-onset POAF did not affect long-term survival after AVR with or without concomitant CABG for symptomatic or asymptomatic severe AS and/or AR.

### 3.8. Presence of Symptoms

The subgroup of patients in which the effect of the presence of symptoms was assessed comprised 562 patients, divided into 50 asymptomatic (NYHA Class I) and 512 symptomatic (NYHA Class II–IV) patients, who underwent AVR with or without concomitant CABG for severe AS.

The number of asymptomatic patients was relatively large. However, because allocation to the different NYHA Classes was done retrospectively, and all of these presumed asymptomatic patients were operated upon, it is possible that some of these presumed asymptomatic patients were in fact slightly symptomatic (NYHA Class II) and were therefore possibly falsely allocated to NYHA Class I instead of Class II.

#### 3.8.1. Baseline Characteristics

As shown in Table 9, the mean age of the symptomatic patients showed a trend toward being higher in comparison with the asymptomatic patients. Operative risk scores in the symptomatic patients were higher than in the asymptomatic patients.

#### 3.8.2. Thirty-Day Mortality

Thirty-day mortality was similar in the patients with symptomatic vs. those with asymptomatic severe AS: 2.9% (*n* = 15) vs. 6.0% (*n* = 3), respectively; *p* = 0.210.

#### 3.8.3. Long-Term Survival

As shown in Figure 11, long-term survival was worse in patients with symptomatic in comparison with those with asymptomatic severe AS: 12.2 ± 0.4 (95% CI: 11.5–13.0) vs. 14.7 ± 1.3 (95% CI: 12.1–17.3) years, respectively; log-rank: *p* = 0.026.

#### 3.8.4. Multivariable Analysis

In Cox proportional hazards analysis, the presence of symptoms was not an independent predictor of decreased long-term survival.

#### 3.8.5. Conclusions

Although the presence of symptoms per se was not an independent predictor of decreased long-term survival after AVR with or without concomitant CABG for severe AS, symptomatic patients with severe AS should be operated upon as soon as possible because the presence of symptoms was significantly associated with both higher operative risk scores and decreased long-term survival. This finding conforms to current American and European guidelines for the management of valvular heart disease, in which AVR is a Class I, Level B, indication for symptomatic severe AS [34,35].

### 3.9. Major Adverse Valve-Related Events

The subgroup of patients in which major adverse valve-related events during long-term follow-up were assessed comprised the total study population of 673 patients, divided into 430 patients with a mechanical and 243 patients with a biologic prosthesis, who underwent AVR with or without concomitant CABG for symptomatic or asymptomatic severe AS and/or AR.

#### 3.9.1. Cumulative Incidence Rates

As shown in Table 10, the cumulative incidence rate of major bleeding events during long-term follow-up was higher in patients with a mechanical in comparison with those with a biologic prosthesis. Related INR values (within 48 h from the events) were similar. The cumulative incidence rate of hemorrhagic stroke events was similar in patients with a mechanical vs. those with a biologic prosthesis, as were the related INR values. The cumulative incidence rate of ischemic stroke events was higher in patients with a biologic in comparison with those with a mechanical prosthesis. Cumulative incidence rates of prosthetic valve endocarditis, permanent pacemaker implantation, and repeat AVR were similar in patients with a mechanical vs. those with a biologic prosthesis.

#### 3.9.2. Major Mechanical Valve-Related Bleeding Events under vs. above the Age of 60

The subgroup of patients with a mechanical prosthesis who had been under vs. those who had been above the age of 60 in which long-term major bleeding and hemorrhagic stroke events were assessed comprised 430 patients, divided into 163 patients under the age of 60 and 267 patients aged 60 years or more at the time of surgery, who had undergone mechanical AVR with or without concomitant CABG for symptomatic or asymptomatic severe AS and/or AR.

#### 3.9.3. Incidence of Major Mechanical Valve-Related Bleeding Events under vs. above the Age of 60

As shown in Table 11, cumulative incidence rates of major bleeding, as well as of hemorrhagic stroke events during long-term follow-up, were similar in patients with a mechanical prosthesis under vs. those above the age of 60. Related INR values were also similar. Related INR values were not exceptionally high. All these findings were also done in a previous study with the same study population as the current study, but with a shorter long-term follow-up (mean 18.1 ± 1.2 years) than in the current study [39].

#### 3.9.4. Conclusions

The cumulative incidence rate of major bleeding events during long-term follow-up after AVR with or without concomitant CABG for symptomatic or asymptomatic severe AS and/or AR was significantly higher in patients with a mechanical in comparison to those with a biologic prosthesis. Younger age (under 60 years) at the time of surgery did not protect patients with a mechanical prosthesis against major bleeding events during long-term follow-up.

## 4. Discussion

Considering the efforts, patience, and costs of infrastructure, very long-term follow-up studies on heart valve surgery are rare. Therefore, we assessed long-term outcomes after AVR, either isolated or with concomitant CABG, during a mean follow-up of 25.1 ± 2.8 years, in a cohort of 673 consecutive patients with symptomatic or asymptomatic severe AS and/or AR. Impaired LV function, severe PPM, and increased ACC time (even within normal limits) were independent predictors of decreased long-term survival. LVH, a mechanical or biologic prosthesis, increased CPB time, new-onset POAF, and the presence of symptoms did not independently predict decreased long-term survival. The risk of major bleeding events during long-term follow-up was higher in patients with a mechanical in comparison with those with a biologic prosthesis. Younger age (under the age of 60) did not protect patients with a mechanical prosthesis against major bleeding events during long-term follow-up. Before undertaking this study, we took the position that within a hospital registry of heart valve patients, the research question can determine the selection of the studied patient group. This requires a strict description of the study group and study methods. In view of the purpose of the current study to focus on the usefulness of very long-term outcomes after AVR in a single institution, we focused on already-known predictors of decreased long-term survival after AVR. Two major conclusions can be drawn. Even in a large-volume hospital, such as the hospital in which the study patients of the current study were operated upon, the number of study patients who are still alive at 15 years of follow-up is quite small (in the current study: 251 of 673 patients) to take the rather low prevalence of valve-related adverse events into consideration. This is even more so when the study population, like the current one, has to be divided into different subgroups to answer specific research questions. Second, with even fewer patients being alive at 25 years of follow-up (in the current study: 87 of 673 patients), it is not to be expected that new risks or surprising conclusions will appear in the period between long and very long-term follow-up. For example, in a previous study with the same study population as the current study but with a shorter long-term follow-up, new-onset POAF was not an independent predictor of decreased long-term survival [38]. In the current study with the same study population as this previous study but with a longer follow-up, new-onset POAF was still not an independent predictor of decreased long-term survival. On the other hand, in a previous study with the same study population as the current study but with a shorter long-term follow-up, severe PPM had only shown a non-significant trend toward decreased long-term survival [36]. In the current study with the same study population as this previous study but with a longer follow-up, severe PPM was indeed an independent predictor of decreased long-term survival. In addition, the results of the current study highlight the dilemmas at the time of surgery 25 years ago, when surgical techniques were less sophisticated than in the present time. Due to more advanced operation techniques, both older patients and those with more comorbidities can today be operated upon with relatively slightly increased operative risks in comparison to the past. To illustrate this, the mean age of the patients of the current study population at the time of surgery between 1990 and 1994 was 65.4 ± 10.7 years. This is in contrast to the mean age of 70.6 ± 10.6 years of the patients in a previous study who had served as a validation set for the development of the “AVR score” and had also undergone AVR with or without concomitant CABG for symptomatic or asymptomatic severe AS and/or AR in the same hospital as the patients of the current study, however two decennia later (between 2007 and 2009) [26]. Partly due to the higher age itself, operative risk scores in the patients operated upon between 2007 and 2009 were higher in comparison to the patients of the current study who had been operated upon between 1990 and 1994 (logistic EuroSCORE: 8.1 ± 7.3 vs. 5.0 ± 5.1, respectively), as were the 30-day mortality rates (3.6% vs. 2.8%, respectively). As the age of the general population continues to rise, and therefore also the age of patients with severe AS, and, partly due to the higher age itself, also their operative risk scores, more patients in the future will be candidates for transcatheter aortic valve implantation, which can today be performed with acceptable operative risks [40]. Apart from improvements in technology and minimally invasive surgery, the findings of the current study reveal that the quality of medical decision-making, especially in terms of timing of the intervention (e.g., performing earlier surgery to prevent impaired LV function), and quality of perioperative care (e.g., treatment of new-onset POAF aimed to restore sinus rhythm within 24 to 48 h from onset, so as soon as possible) can determine not only short-term but also very long-term survival. Additionally, using sutureless or rapid-deployment valves can improve valve size by the absence of a sewing ring (preventing severe PPM) and can also reduce ACC and CBP times (striving for as short ACC and CPB times as possible) [41]. Moreover, the current study confirms that the surgical team has to look further than the parameters determining operative mortality as calculated by EuroSCORE, EuroSCORE II, or STS-PROM because these operative risk scores were developed and validated only to predict mortality on in the very short run [42,43,44]. As shown in the current study, prognostic factors associated with the implantation, such as the timing of surgery, preoperative LV function, and prosthesis size, potentially have a negative very long-term impact, which should be addressed at the time of surgery. However, these conclusions are made for a routine cardiac surgical intervention mainly applied in elderly patients. In comparison with heart valve surgery in the elderly, very long-term follow-up studies have been more frequently conducted in patients who underwent cardiac surgery for congenital disease [45]. The feedback of very long-term follow-up in these adult patients with congenital heart disease had a substantial impact on the timing and type of interventions [46]. Very long-term follow-up studies also provide a standard for comparing results of combinations of percutaneous and conventional treatments [47]. The connection with the initial decision-making on the indication for heart valve surgery and operating team usually fades with new generations. Especially, as shown above, when the target group exists of elderly patients similar to the study population of the current study, rather few cases remain during very long-term follow-up to draw conclusions about observed major adverse valve-related events and interpretation of operative risks. However, as shown in the current study on the subjects “new-onset POAF” and “major bleeding events” in patients with a mechanical prosthesis under vs. those above the age of 60, very long-term follow-up studies can reveal that findings of medium-term follow-up indeed sufficiently confirm the future course in terms of life expectancy. It is not surprising that in a younger population of patients who underwent mitral valve repair surgery, very long-term follow-up revealed interesting findings. It showed that in the very long run, the closed mitral annulus ring device was associated with serious mitral valve stenosis [48]. This study suggested that the surgeon should choose an open or closed annular reduction technique depending on the patient’s valve anatomy. Choices of device and surgical practices and decision-making at the time of surgery may have a long-term impact despite appropriate aftercare. This implies that it is worthwhile that hospitals review the quality of heart valve surgery on a regular basis. This may be supported by the existence of large international and national registries, such as the National Heart Registration (NHR) registry in The Netherlands [49]. However, very long-term follow-up studies seem feasible and useful only when sufficient patients are included with an aimed life expectancy that matches the ambition of the duration of follow-up. In the current study, only 2 of 243 patients of the total study population with a biological prosthesis (mean age at the time of surgery: 74.7 ± 4.6 years) remained available at 25 years of follow-up, as at the time of surgery, patients were generally considered eligible for a biologic AVR at an age of 75 years or older. Today, implantation of a biological prosthesis in patients 60 years of age or younger is common, and risks of redo surgery or transcatheter valve-in-valve implantation are accepted bail-out strategies in cases of prosthetic valve failure [50].

## 5. Conclusions

Despite technology-induced improvements, there remain many reasons to learn from the past. This type of learning from experience should be supported by long-term and very long-term follow-up of valve surgery cohorts, preferably younger patients in multiple centers. It is recommended that the profession convinces national and international registries to follow well-described cohorts for a very long term.

## Figures and Tables

**Figure 1 jcm-10-03925-f001:**
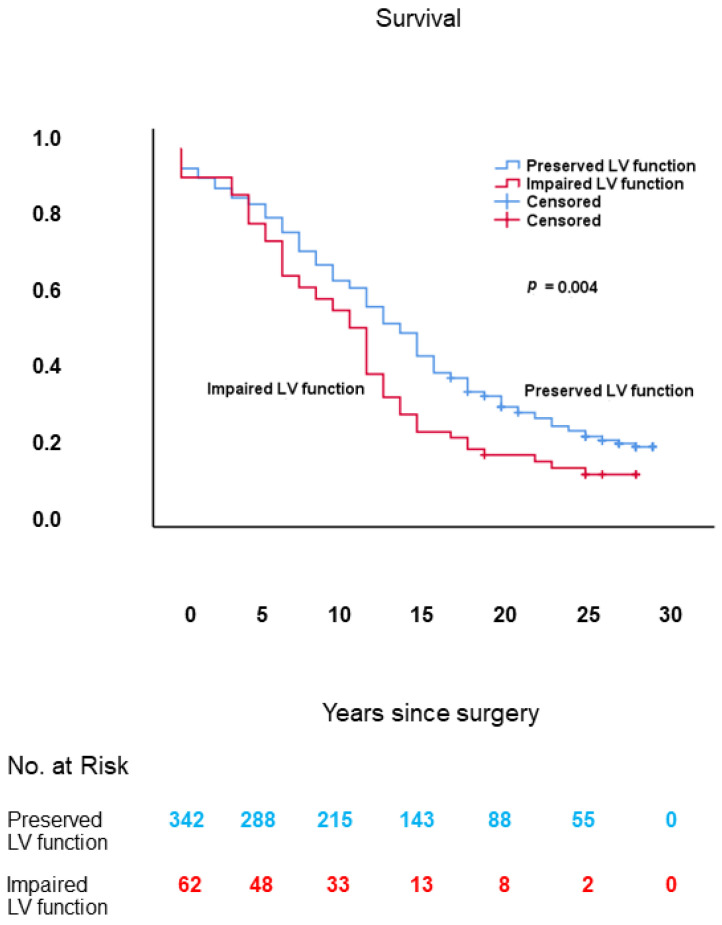
Kaplan–Meier survival curves, from date of surgery, of patients with impaired vs. those with preserved LV function. LV: left ventricular.

**Figure 2 jcm-10-03925-f002:**
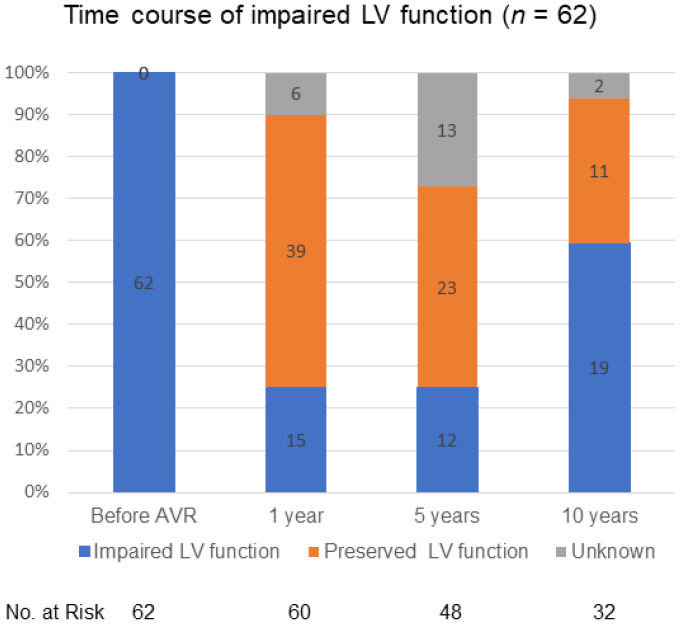
Time course of impaired LV function during 10 years of follow-up. LV: left ventricular. AVR: aortic valve replacement.

**Figure 3 jcm-10-03925-f003:**
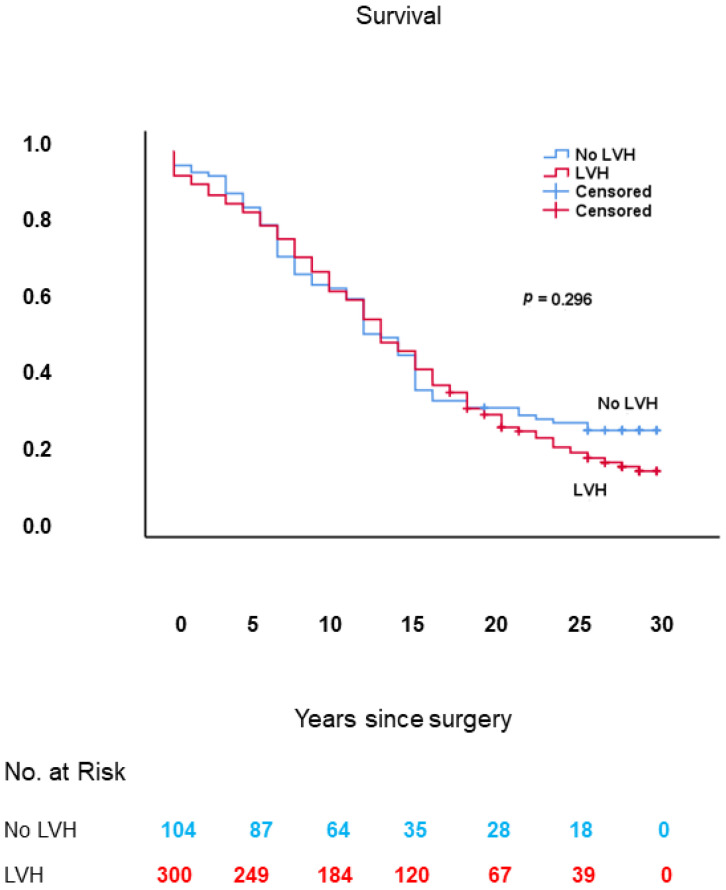
Kaplan–Meier survival curves, from date of surgery, of patients with vs. those without LVH. LVH: left ventricular hypertrophy.

**Figure 4 jcm-10-03925-f004:**
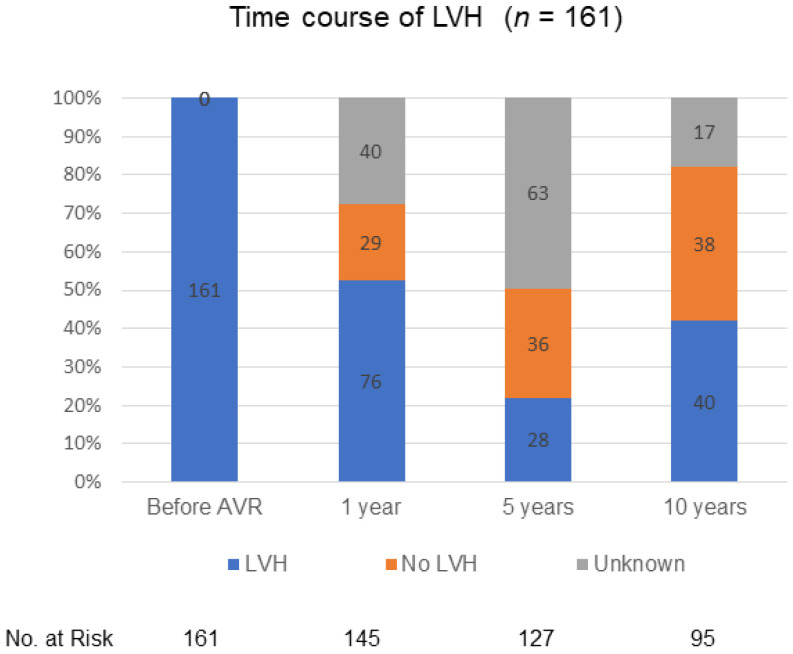
Time course of LVH during 10 years of follow-up, after excluding the patients with a history of hypertension. LVH: left ventricular hypertrophy. AVR: aortic valve replacement.

**Figure 5 jcm-10-03925-f005:**
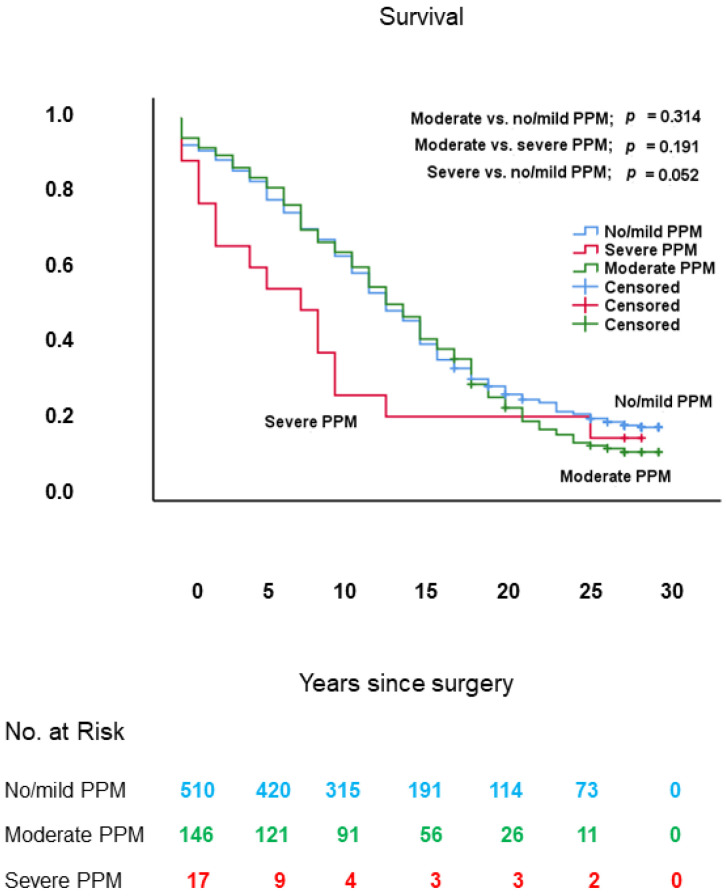
Kaplan–Meier survival curves, from date of surgery, of patients with moderate PPM, severe PPM, and no/mild PPM. PPM: prosthesis–patient mismatch.

**Figure 6 jcm-10-03925-f006:**
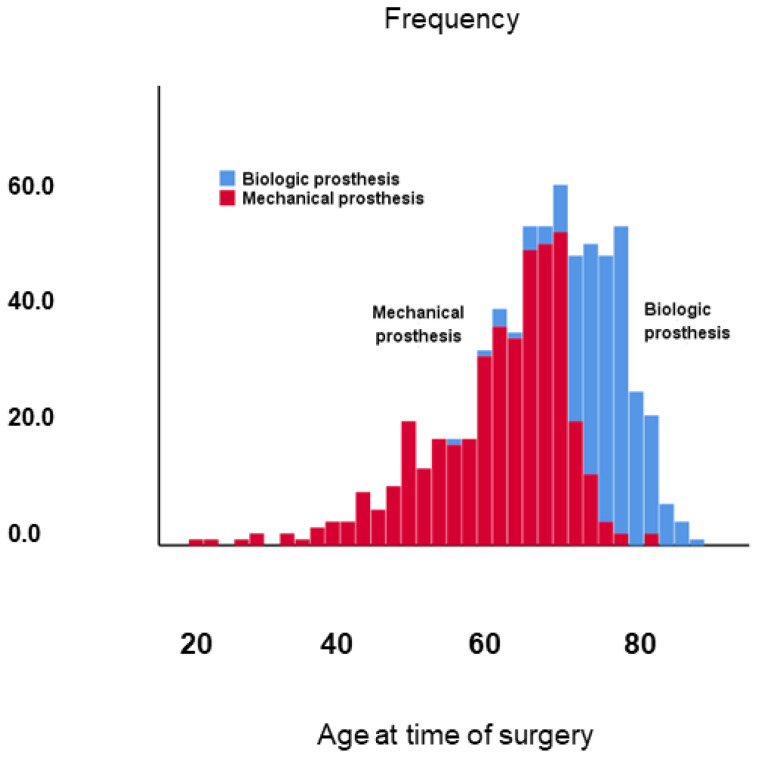
Histogram showing the frequency of patients with a mechanical vs. those with a biologic prosthesis in the total study population of 673 patients.

**Figure 7 jcm-10-03925-f007:**
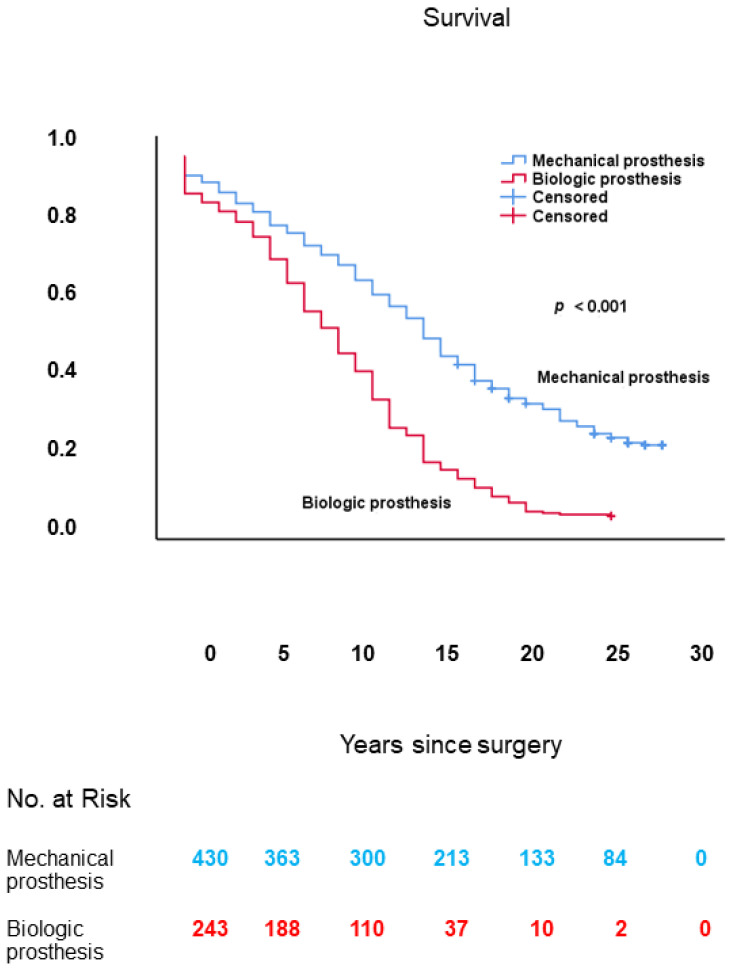
Kaplan–Meier survival curves of patients with a biologic vs. those with a mechanical aortic valve prosthesis.

**Figure 8 jcm-10-03925-f008:**
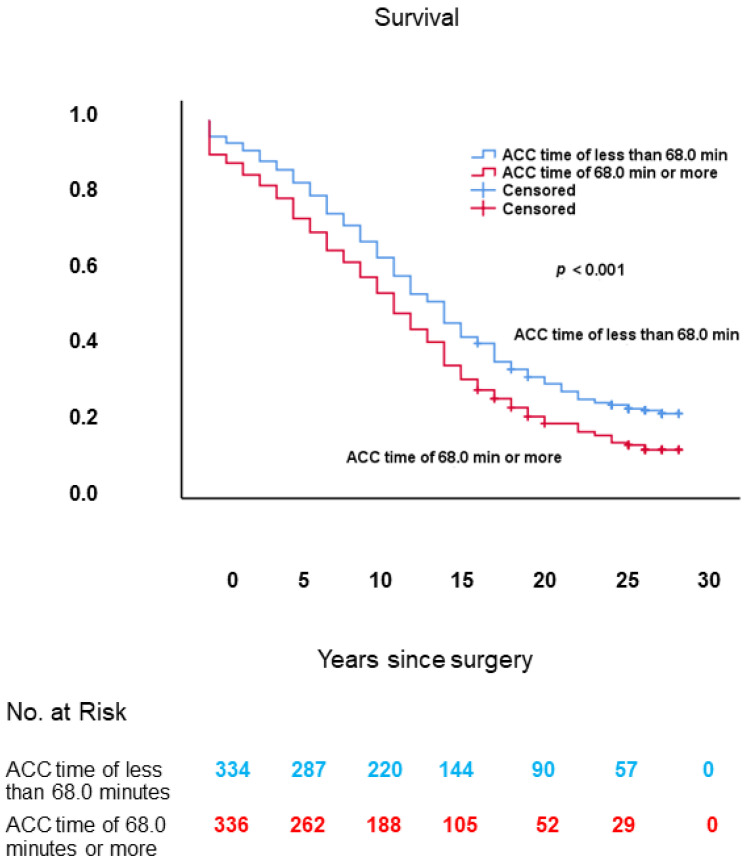
Kaplan–Meier survival curves, from date of surgery, of patients with ACC time of 68.0 min or more vs. those with ACC time of less than 68.0 min. ACC: aortic cross-clamp.

**Figure 9 jcm-10-03925-f009:**
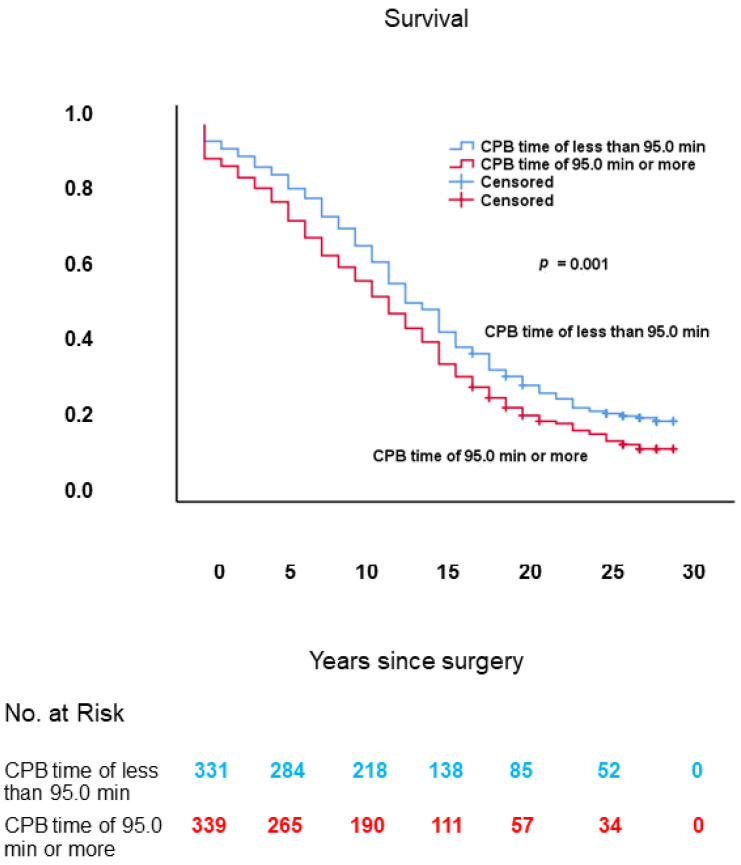
Kaplan–Meier survival curves, from date of surgery, of patients with CPB time of 95.0 min or more vs. those with CPB time of less than 95.0 min. CPB: cardiopulmonary bypass.

**Figure 10 jcm-10-03925-f010:**
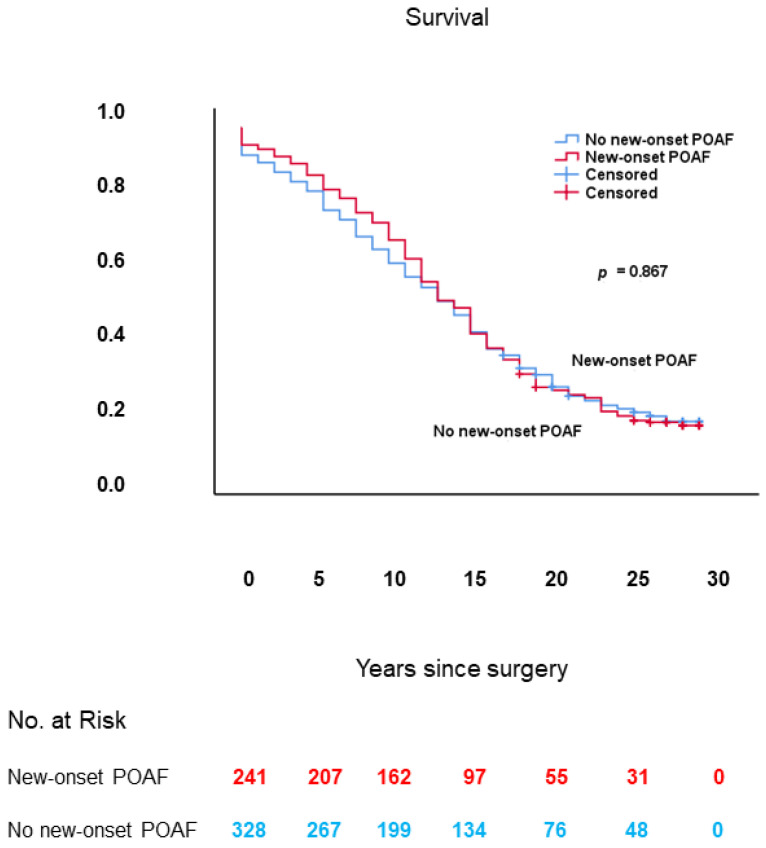
Kaplan–Meier survival curves, from date of surgery, of patients with new-onset POAF vs. those without. POAF: postoperative atrial fibrillation.

**Figure 11 jcm-10-03925-f011:**
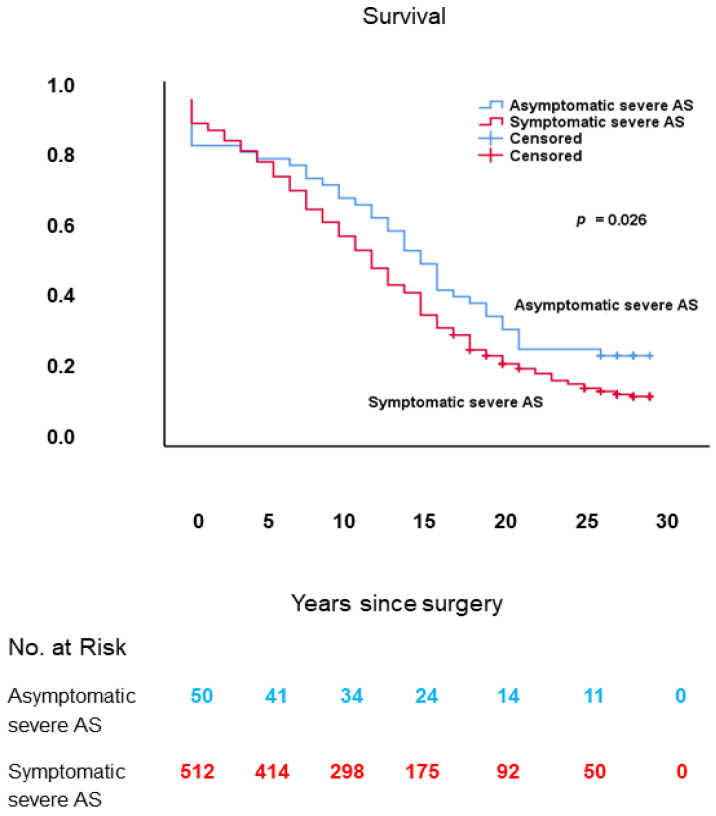
Kaplan–Meier survival curves, from date of surgery, of patients with symptomatic vs. those with asymptomatic severe AS. AS: aortic stenosis.

**Table 1 jcm-10-03925-t001:** Baseline characteristics.

Characteristic	Impaired LV ^1^ Function (*n* = 62)	Preserved LV ^1^ Function (*n* = 342)	*p*-Value
Age (years)	65.3 ± 9.6	64.9 ± 11.0	0.794
Logistic EuroSCORE ^2^	7.8 ± 5.2	3.9 ± 2.9	<0.001
EuroSCORE II ^2^	2.9 ± 2.2	1.6 ± 1.6	<0.001
STS PROM ^3^	2.4 ± 1.8	1.6 ± 1.4	<0.001

Values are presented as mean ± standard deviation. ^1^ LV: left ventricular; ^2^ EuroSCORE: European System for Cardiac Operative Risk Evaluation; ^3^ STS PROM: Society of Thoracic Surgeons Predicted Risk of Mortality.

**Table 2 jcm-10-03925-t002:** Baseline characteristics.

Characteristic	LVH ^1^ (*n* = 300)	No LVH ^1^ (*n* = 104)	*p*-Value
Age (years)	66.1 ± 10.1	61.7 ± 12.0	<0.001
Logistic EuroSCORE ^2^	4.5 ± 3.5	4.6 ± 4.1	0.737
EuroSCORE II ^2^	1.7 ± 1.8	1.8 ± 1.7	0.685
STS PROM ^3^	1.7 ± 1.4	1.8 ± 1.8	0.359

Values are presented as mean ± standard deviation. ^1^ LVH: left ventricular hypertrophy; ^2^ EuroSCORE: European System for Cardiac Operative Risk Evaluation; ^3^ STS PROM: Society of Thoracic Surgeons Predicted Risk of Mortality.

**Table 3 jcm-10-03925-t003:** Baseline characteristics of the patients with PPM ^1^ vs. those with no/mild PPM ^1^.

Characteristic	PPM ^1^ (*n* = 163)	No/mild PPM ^1^ (*n* = 510)	*p*-Value
Age (years)	68.3 ± 9.6	64.4 ± 10.9	<0.001
Logistic EuroSCORE ^2^	5.6 ± 4.5	4.8 ± 5.2	0.094
EuroSCORE II ^2^	2.6 ± 2.5	2.2 ± 2.6	0.049
STS PROM ^3^	2.1 ± 1.5	1.9 ± 1.6	0.274

Values are presented as mean ± standard deviation. ^1^ PPM: prosthesis–patient mismatch; ^2^ EuroSCORE: European System for Cardiac Operative Risk Evaluation; ^3^ STS PROM: Society of Thoracic Surgeons Predicted Risk of Mortality.

**Table 4 jcm-10-03925-t004:** Baseline characteristics of the patients with severe PPM vs. those with moderate PPM.

Characteristic	Severe PPM ^1^ (*n* = 17)	Moderate PPM ^1^ (*n* = 146)	*p*-Value
Age (years)	69.9 ± 10.0	68.1 ± 9.6	0.453
Logistic EuroSCORE ^2^	6.5 ± 6.6	5.5 ± 4.3	0.400
EuroSCORE II ^2^	2.7 ± 2.4	2.6 ± 2.6	0.960
STS PROM ^3^	2.2 ± 1.2	2.1 ± 1.6	0.704

Values are presented as mean ± standard deviation. ^1^ PPM: prosthesis–patient mismatch; ^2^ EuroSCORE: European System for Cardiac Operative Risk Evaluation; ^3^ STS PROM: Society of Thoracic Surgeons Predicted Risk of Mortality.

**Table 5 jcm-10-03925-t005:** Baseline characteristics.

Characteristic	Mechanical AVR ^1^ (*n* = 430)	Biologic AVR ^1^ (*n* = 243)	*p*-Value
Age (years)	60.1 ± 9.5	74.7 ± 4.6	<0.001
Logistic EuroSCORE ^2^	3.8 ± 5.0	7.0 ± 4.5	<0.001
EuroSCORE II ^2^	1.9 ± 2.2	3.0 ± 3.0	<0.001
STS PROM ^3^	1.5 ± 1.4	2.7 ± 1.7	<0.001

Values are presented as mean ± standard deviation. ^1^ AVR: aortic valve replacement; ^2^ EuroSCORE: European System for Cardiac Operative Risk Evaluation; ^3^ STS PROM: Society of Thoracic Surgeons Predicted Risk of Mortality.

**Table 6 jcm-10-03925-t006:** Baseline characteristics of patients with ACC time of less than 68.0 min vs. 68.0 min or more.

Characteristic	ACC ^1^ Time < 68.0 min (*n* = 334)	ACC ^1^ Time ≥ 68.0 min (*n* = 336)	*p*-Value
Age (years)	64.6 ± 11.3	66.1 ± 10.1	0.059
Logistic EuroSCORE ^2^	4.4 ± 3.6	5.6 ± 6.2	0.001
EuroSCORE II ^2^	1.7 ± 1.7	2.9 ± 3.2	<0.001
STS PROM ^3^	1.7 ± 1.5	2.2 ± 1.7	<0.001

Values are presented as mean ± standard deviation. ^1^ ACC: aortic cross-clamp; ^2^ EuroSCORE: European System for Cardiac Operative Risk Evaluation; ^3^ STS PROM: Society of Thoracic Surgeons Predicted Risk of Mortality.

**Table 7 jcm-10-03925-t007:** Baseline characteristics of patients with CPB time of less than 95.0 min vs. 95.0 min or more.

Characteristic	CPB ^1^ Time < 95.0 min (*n* = 331)	CPB ^1^ Time ≥ 95.0 min (*n* = 339)	*p*-Value
Age (years)	64.4 ± 11.2	66.2 ± 10.1	0.031
Logistic EuroSCORE ^2^	4.2 ± 3.2	5.8 ± 6.3	<0.001
EuroSCORE II ^2^	1.6 ± 1.3	2.9 ± 3.3	<0.001
STS PROM ^3^	1.6 ± 1.4	2.2 ± 1.8	<0.001

Values are presented as mean ± standard deviation. ^1^ CPB: cardiopulmonary bypass; ^2^ EuroSCORE: European System for Cardiac Operative Risk Evaluation; ^3^ STS PROM: Society of Thoracic Surgeons Predicted Risk of Mortality.

**Table 8 jcm-10-03925-t008:** Baseline characteristics.

Characteristic	New-Onset POAF ^1^ (*n* = 241)	No New-Onset POAF (*n* = 328)	*p*-Value
Age (years)	65.4 ± 10.7	64.1 ± 10.9	0.158
Logistic EuroSCORE ^2^	4.7 ± 3.8	4.9 ± 5.9	0.747
EuroSCORE II ^2^	2.2 ± 2.1	2.2 ± 3.0	0.995
STS PROM ^3^	1.8 ± 1.3	1.7 ± 1.6	0.635

Values are presented as mean ± standard deviation. ^1^ POAF: postoperative atrial fibrillation; ^2^ EuroSCORE: European System for Cardiac Operative Risk Evaluation; ^3^ STS PROM: Society of Thoracic Surgeons Predicted Risk of Mortality.

**Table 9 jcm-10-03925-t009:** Baseline characteristics.

Characteristic	Symptomatic Severe AS ^1^ (*n* = 512)	Asymptomatic Severe AS ^1^ (*n* = 50)	*p*-Value
Age (years)	67.2 ± 9.3	64.7 ± 10.5	0.073
Logistic EuroSCORE ^2^	5.2 ± 4.4	3.8 ± 3.5	0.023
EuroSCORE II ^2^	2.4 ± 2.6	1.3 ± 1.4	<0.001
STS PROM ^3^	2.0 ± 1.5	1.3 ± 0.9	<0.001

Values are presented as mean ± standard deviation. ^1^ AS: aortic stenosis; ^2^ EuroSCORE: European System for Cardiac Operative Risk Evaluation; ^3^ STS PROM: Society of Thoracic Surgeons Predicted Risk of Mortality.

**Table 10 jcm-10-03925-t010:** Major adverse valve-related events during long-term follow-up.

Major Adverse Valve-Related Event	Mechanical Prosthesis (*n* = 430)	Biologic Prosthesis (*n* = 243)	*p*-Value
Major bleeding	132 (30.7)	46 (18.9)	0.001
Related INR ^1^	4.2 ± 1.7	4.3 ± 2.1	0.838
Range	1.2–9.2	1.2–9.7	
Hemorrhagic stroke	36 (8.4)	14 (5.8)	0.284
Related INR ^1^	3.7 ± 1.0	6.0 ± 5.3	0.345
Range	2.2–7.5	1.4–15.0	
Ischemic stroke	34 (7.9)	34 (14.0)	0.016
Prosthetic valve endocarditis	18 (4.2)	13 (5.3)	0.566
Permanent pacemaker implantation	37 (8.6)	19 (7.8)	0.773
Repeat aortic valve replacement	20 (4.7)	9 (3.7)	0.694

Values are presented as *n* (%) or as mean ± standard deviation. ^1^ INR: international normalized ratio.

**Table 11 jcm-10-03925-t011:** Major mechanical valve-related bleeding events under vs. above the age of 60.

Major Mechanical Valve-Related Bleeding Event	Age < 60 Years (*n* = 163)	Age ≥ 60 Years (*n* = 267)	*p*-Value
Major bleeding	48 (29.4)	84 (31.5)	0.747
Related INR ^1^	4.1 ± 1.2	4.3 ± 2.0	0.628
Range	2.3–7.5	1.2–9.2	
Hemorrhagic stroke	16 (9.8)	20 (7.5)	0.473
Related INR	3.5 ± 0.8	4.0 ± 1.3	0.279
Range	2.2–4.8	2.8–7.5	

Values are presented as *n* (%) or as mean ± standard deviation. ^1^ INR: international normalized ratio.

## Data Availability

The data associated with the paper are not publicly available but are available from the corresponding author on request.

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
