# Peer review of "What Can We Learn from the Past by Means of Very Long-Term Follow-Up after Aortic Valve Replacement?"

_jcm, 2021, doi:10.3390/jcm10173925_

Round 1

Reviewer 1 Report

impressive long-term follow-up of 25.1 ± 2.8 years. Predictors of decreased long-term survival, cumulative incidence rates of major adverse events in patients with a mechanical versus those with a biological valves, major bleeding events under and above the age of 60 were determined in their extensively performed study. The result section is presented uncommonly. However, there might be no other option due to the enormous wealth of generated data. The discussion is very well-written and structured. In total, the manuscript is of great interest of the Journal’s Reader.

Still, the Authors should address some minor questions:

  1. It is surprising that moderate PPM does not affect the long-term survival. We obviously have to take the data as it is. However, this finding is more than surprising and not in accordance with the current literature. A further statement in the already performed PPM discussion is needed.
  2. The amount of asymptomatic operated patients is surprisingly high in the described cohort. What is your possible explanation for that?
  3. Is there any data on the implanted valves (manufacturer & valve sizes) available? The impact might be of interest, however this data would scope the result and discussion section.

Author Response

Reviewer 1

Remark 1:

“impressive long-term follow-up of 25.1 ± 2.8 years. Predictors of decreased long-term survival, cumulative incidence rates of major adverse events in patients with a mechanical versus those with a biological valves, major bleeding events under and above the age of 60 were determined in their extensively performed study. The result section is presented uncommonly. However, there might be no other option due to the enormous wealth of generated data. The discussion is very well-written and structured. In total, the manuscript is of great interest of the Journal’s Reader.”

Answer 1:

We thank Reviewer 1 for these words. Indeed the results of our study were presented uncommonly, but by our opinion it was the only way to present the data as structured as possible.

Changes 1:

We would like to suggest to make no changes in the revised manuscript regarding this issue.

Remark 2:

“It is surprising that moderate PPM does not affect the long-term survival. We obviously have to take the data as it is. However, this finding is more than surprising and not in accordance with the current literature. A further statement in the already performed PPM discussion is needed.”

Answer 2:

We agree with Reviewer 1 that it was surprising that moderate prosthesis-patient mismatch (PPM) did not affect long-term survival. Although some previous studies have also failed to demonstrate that moderate PPM was a significant predictor of decreased long-term survival after aortic valve replacement (e.g. “Daneshvar, S.A.; Rahimtoola, S.H. Valve prosthesis-patient mismatch [VP-PM]: a long-term perspective. J Am Coll Cardiol 2012, 60, 1123-1135; DOI: 10.1016/j.jacc.2012.05.035”), other studies indeed have shown that both moderate and severe PPM were indeed independent predictors of decreased long-term survival (e.g. “Head, S.J.; Mokhles, M.M.; Osnabrugge, R.L.; Pibarot, P.; Mack, M.J.; Takkenberg, J.J.; Bogers, A.J.; Kappetein, A.P. The impact of prosthesis-patient mismatch on long-term survival after aortic valve replacement: a systematic review and meta-analysis of 34 observational studies comprising 27 186 patients with 133141 patient-years. Eur Heart J 2012, 33, 1518-1529; DOI:10.1093/eurheartj/ehs003”, or “Sá, M.P.B.O.; de Carvalho, M.M.B.; Sobral Filho, D.C.; Cavalcanti, L.R.P.; Rayol, S.D.C.; Diniz, R.G.S.; Menezes, A.M.; Clavel, M.A.; Pibarot, P.; Lima, R.C. Surgical aortic valve replacement and patient-prosthesis mismatch: a meta-analysis of 108 182 patients. Eur J Cardiothorac Surg 2019, 56, 44-54; DOI:10.1093/ejcts/ezy466”). The finding that moderate PPM was not an independent predictor of decreased long-term survival, and even showed a non-significant trend toward an increased, not a decreased, long-term survival in comparison with no/mild PPM (hazards ratio 0.856 [95% confidence interval: 0.685 – 1.071]; p = 0.174) was difficult to explain because patients with moderate PPM had not been younger and their operative risk scores had not been lower in comparison with those with no/mild PPM. We also found this inverse survival relationship between moderate PPM and long-term survival in our previous paper on this subject with the same study population, but with a shorter (mean 17.8 ± 1.8 years) long-term follow up (reference number 36 of both the original and the revised manuscript): “Swinkels, B.M.; de Mol, B.A.; Kelder, J.C.; Vermeulen, F.E.; Ten Berg, J.M. Prosthesis-patient mismatch after aortic valve replacement: effect on long-term survival. Ann Thorac Surg 2016, 101, 1388-1394; DOI:10.1016/j.athoracsur.2016.01.048). As described in that previous paper, this inverse survival relationship could also not be explained statistically. So was the used Cox regression model at least reasonable and after putting other variables into the model, such as STS-PROM operative risk score, this inverse survival relationship was not essentially changed. Also, using the indexed effective orifice area as a continuous variable, both linearly and nonlinearly, or stratifying by age group, or using age as a nonlinear variable, did not essentially change this inverse survival relationship. Looking only at the first year of follow-up, this inverse survival relationship was also retained.

Changes 2:

On page 11, line 354, of the revised manuscript, we discussed this issue of the inverse survival relationship between moderate PPM and long-term survival.

Remark 3:

“The amount of asymptomatic operated patients is surprisingly high in the described cohort. What is your possible explanation for that?

Answer 3:

We agree with Reviewer 1 that the amount of asymptomatic (New York Heart Association [NYHA] Class I), but still operated patients with severe aortic stenosis (50 of 562 [8.9%]) was surprisingly high. However, because allocation to the different NYHA Classes was done retrospectively, and all of these presumed asymptomatic patients were operated upon, it is possible that some of these presumed asymptomatic patients were in fact slightly symptomatic (NYHA Class II) and were therefore possibly falsely allocated to NYHA Class I instead of Class II. The main massage of our study on this subject was to show that long-term survival of symptomatic patients (taken patients in NYHA Class II to IV together) was worse in comparison with asymptomatic patients (those in NYHA Class I). Although the presence of symptoms per se was not an independent predictor of decreased long-term survival, we think that symptomatic patients with severe aortic stenosis should be operated upon as soon as possible because the presence of symptoms was significantly associated with both higher operative risk scores and decreased long-term survival.

Changes 3:

In the Methods section on page 3, line 138, we defined asymptomatic patients as those who had been in NYHA Class I and asymptomatic patients as those who had been in Class II to IV. Also, on page 19, line 548 of the revised manuscript, we discussed the possibility that in fact less than the 50 presumed asymptomatic patients with severe aortic stenosis had been in NYHA Class I.

Remark 4:

“Is there any data on the implanted valves (manufacturer & valve sizes) available? The impact might be of interest, however this data would scope the result and discussion section.”

Answer 4:
We agree with Reviewer 1 that we did not present data on manufacturer and valve sizes. In the 673 study patients, the following mechanical (n = 430) and biologic (n = 243) prostheses were implanted. Mechanical prostheses: St. Jude Medical Standard (n = 205), St. Jude Medical Hemodynamic Plus (n = 3), Sorin Allcarbon (n = 211, including two 33 mm sized Sorin Allcarbon mitral valve mechanical prostheses, placed upside down in the aortic position), and the Sorin Bicarbon mechanical prosthesis (n = 11). Biologic prostheses: Medtronic Intact (n = 100), CE-SAV (n = 139), and CE Perimount biologic prosthesis (n = 4). The most frequent prosthesis size among the mechanical prostheses was 23 mm (n = 128), followed by 25 mm (n = 113). The most frequent prosthesis size among the biologic prostheses was also 23 mm (n = 101), followed by 25 mm (n = 80). PPM occurred in 88 of 430 (20.5%) patients with a mechanical prosthesis and in 75 of 243 (30.9%) patients with a biologic prosthesis. Severe PPM occurred in 8 of 430 (1.9%) patients with a mechanical and in 9 of 243 (3.3%) patients with a biologic prosthesis. PPM occurred only in patients with a mechanical prosthesis sized 23 mm or smaller or with a biologic prosthesis sized 25 mm or smaller. Severe PPM occurred only in patients with a mechanical prosthesis sized 23 mm or smaller or with a biologic prosthesis sized 23 mm.

Changes 4:

We added the abovementioned data on manufacturer and valve sizes in the “PPM section” on page 9, line 304, and in the “Mechanical versus biologic prostheses section” on page 12, line 377 of the revised manuscript.

Reviewer 2 Report

The Authors aim at describing the findings of the retrospective analysis of a single-center experience on surgical AVR with a long-term follow-up of 25 years. The manuscript is well written and the complexity of the results has been made easier breaking down the data on each endpoint. The Authors give important and useful information, but most of all reinforce the idea that the currently used operative risk scores should be modified to include other aspects regarding patients’ clinical data. The paper is overall lengthy and would benefit from a more concise introduction, description of the study population and final discussion. This study presents several limits: it’s retrospective, single-center and, as the Authors themselves highlight, refers to a time when surgical techniques and equipment were not as sophisticated as the ones available today. Yet, it couldn’t be any different aiming at such a wide timespan for the follow-up.

Just a few remarks:

  • May the Authors explain why they put as a reference for long-term outcome of heart valve treatments a publication regarding coronary artery bypass grafting (ref. 11, page 1, line 38)?
  • Page 1, line 44. “[…], and so on” seems inappropriate in this sentence; I would suggest to replace with “and others”.

Author Response

Reviewer 2

Remark 1:

“The Authors aim at describing the findings of the retrospective analysis of a single-center experience on surgical AVR with a long-term follow-up of 25 years. The manuscript is well written and the complexity of the results has been made easier breaking down the data on each endpoint. The Authors give important and useful information, but most of all reinforce the idea that the currently used operative risk scores should be modified to include other aspects regarding patients’ clinical data. The paper is overall lengthy and would benefit from a more concise introduction, description of the study population and final discussion. This study presents several limits: it’s retrospective, single-center and, as the Authors themselves highlight, refers to a time when surgical techniques and equipment were not as sophisticated as the ones available today. Yet, it couldn’t be any different aiming at such a wide timespan for the follow-up.”

Answer 2:

We thank Reviewer 2 for these words. Indeed, the currently used operative risk scores only predict short-term mortality. We discussed this matter already on page 23, line 686, of the “Discussion section” of the original manuscript, stating that “the current study confirmed that the surgical team has to look further than the parameters determining operative mortality as calculated by EuroSCORE, EuroSCORE II, or STS-PROM, because these operative risk scores were developed and validated only to predict mortality in the very short run.” We agree with Reviewer 1 that the manuscript was overall lenghty and would benefit from a more concise introduction, description of the study population and final discussion. However, we think it is essential for a thorough understanding of the different research questions to describe the different study populations in detail, almost inevitably leading to a lenghty paper.Also, we think that the paper would be relatively incomplete when some subjects would have been deleted. In the current study, very long-term follow-up data were sometimes not (e.g. new-onset postoperative atrial fibrillation), but were sometimes indeed (e.g. prosthesis-patient mismatch) invaluable for careful decision making on the indication for aortic valve replacement, both in the “Heart Team” and in the outpatient clinic during patient counseling. The retrospective design indeed had its inherent drawbacks on inclusion and follow-up of study patients. However, performing a strictly prospective study with 25 years of follow-up would have been challenging.

Changes 1:

We would like to propose to not shorten the text of the current paper to prevent confusions as described above.

Remark 2:

“May the Authors explain why they put as a reference for long-term outcome of heart valve treatments a publication regarding coronary artery bypass grafting (ref. 11, page 1, line 38)?”

Answer 2:

We want to thank Reviewer 2 for this critical remark because indeed this publication on quality of life during long-term follow-up applied only to patients who had undergone coronary artery bypass grafting. Therefore, we want to apologize for this mistake.

Changes 2:

On page 25, line 780, of the revised manuscript, we replaced the wrong reference nr. 11 by a systematic review on quality of life during long-term follow-up after aortic valve replacement by a study of Shan et al.

Remark 3:

“Page 1, line 44. “[…], and so on” seems inappropriate in this sentence; I would suggest to replace with “and others.”

Answer 3:

We agree with Reviewer 2 that the words “and others” would have been more appropriate than “and so on”.

Changes 3:

On page 1, line 44, of the revised manuscript, we replaced the words “and so on” by the words “and others”.